# Differential cell-intrinsic regulations of germinal center B and T cells by miR-146a and miR-146b

Sunglim Cho [1], Hyang-Mi Lee[1], I-Shing Yu [2], Youn Soo Choi[3,4], Hsi-Yuan Huang[5], Somaye Sadat Hashemifar[6], Ling-Li Lin[1], Mei-Chi Chen[1], Nikita D. Afanasiev[1], Aly Azeem Khan[6], Shu-Wha Lin[7,8], Alexander Y. Rudensky[9], Shane Crotty [3,10] & Li-Fan Lu [1,11,12]

Reciprocal interactions between B and follicular T helper (Tfh) cells orchestrate the germinal center (GC) reaction, a hallmark of humoral immunity. Abnormal GC responses could lead to the production of pathogenic autoantibodies and the development of autoimmunity. Here we show that miR-146a controls GC responses by targeting multiple CD40 signaling pathway components in B cells; by contrast, loss of miR-146a in T cells does not alter humoral responses. However, specific deletion of both miR-146a and its paralog, miR-146b, in T cells increases Tfh cell numbers and enhanced GC reactions. Thus, our data reveal differential cell-intrinsic regulations of GC B and Tfh cells by miR-146a and miR-146b. Together, members of the miR-146 family serve as crucial molecular brakes to coordinately control GC reactions to generate protective humoral responses without eliciting unwanted autoimmunity.

[1] Division of Biological Sciences, University of California, La Jolla, San Diego, CA 92093, USA. [2] Laboratory Animal Center, College of Medicine, National Taiwan University, Taipei 100, Taiwan. [3] Division of Vaccine Discovery, La Jolla Institute for Allergy and Immunology, La Jolla, CA 92037, USA. [4] Department of Medicine, College of Medicine, Seoul National University, Seoul 03080, Korea. [5] Department of Laboratory Medicine, China Medical University Hospital, China Medical University, Taichung, Taiwan. [6] Toyota Technological Institute, Chicago, IL 60637, USA. [7] Department of Clinical Laboratory Sciences and Medical Biotechnology, College of Medicine, National Taiwan University, Taipei 100, Taiwan. [8] Department of Laboratory Medicine, National Taiwan University Hospital, Taipei 100, Taiwan. [9] Howard Hughes Medical Institute and Immunology Program, Ludwig Center at Memorial Sloan-Kettering Cancer Center, Memorial Sloan-Kettering Cancer Center, New York, NY 10065, USA. [10] Division of Infectious Diseases, Department of Medicine, University of California, La Jolla, San Diego, CA 92037, USA. [11] Moores Cancer Center, University of California, La Jolla, San Diego, CA 92093, USA. [12] Center for Microbiome Innovation, University of California, La Jolla, San Diego, CA 92093, USA. These authors contributed equally: Sunglim Cho, Hyang-Mi Lee. Correspondence and requests for materials should be addressed to L.-F.L. (email: lifanlu@ucsd.edu)

To combat enormously diverse microbial pathogens, different cellular and molecular players need to work in cooperation with, or sometimes in opposition to each other to generate effective immunity. When first-line innate immune responses fail to control infection, B and T cells work in synergy to mount humoral and cellular adaptive immune responses. In the absence of B cells, T cells display poor priming and impaired clonal expansion upon antigen stimulation[1]. Likewise, absent T cells, mice fail to develop germinal centers (GCs), where memory B cells and high affinity antibody-producing plasma cells are generated[2]. The reciprocal dependency between these two major immune cell subsets has become even more evident with the discovery of a specialized T cell subset known as follicular helper T (Tfh) cells[3]. Tfh cells express elevated levels of the chemokine receptor CXCR5, which allows them to respond to CXCL13 and migrate into B cell follicles. The colocalization and interaction between Tfh and B cells are crucial for the induction of the GC reaction, the production of specialized, high affinity antibodies, and the generation of long-term protective immunity. The identification of transcription repressor Bcl6 as a key transcription factor in Tfh cell differentiation has further substantiated the notion that Tfh cells comprise a distinct T helper cell lineage similar to Th1, Th2, Th17, or regulatory T (Treg) cells[3]. Interestingly, Bcl6 was originally identified as an essential regulator of GC B cell differentiation[4]. The fact that Bcl6 controls the development of both GC B cells and Tfh cells suggested a common gene regulatory circuit can be implemented in different immune populations to enable them to play their specialized roles in producing a concerted response to a particular environmental stimulus.

Like Bcl6, microRNA (miR)-146a was shown to be highly expressed in both Tfh and GC B cells[5]. Recent studies have showed that miR-146a plays a prominent role in different aspects of immune cell biology[6]. Both Toll-like receptor (TLR) signaling and Th1 cytokines can strongly upregulate miR-146a expression levels in myeloid cells and Th1 cells, respectively[7,8]. In turn, miR-146a limits the activation and the function of the aforementioned immune cells through repressing key molecules downstream of the corresponding signaling pathways[7,9]. Considering the fact that dysregulated humoral immune responses and heightened production of autoantibodies have been previously reported in mice devoid of miR-146a[10,11], it is thus conceivable that the elevated levels of miR-146a detected in both Tfh and GC B cells might also be essential to restrain the responses of these two immune cell populations. Indeed, two recent studies have implicated miR-146a as a negative regulator of Tfh cell responses[11,12]. Specifically, it was suggested that miR-146a was able to limit the accumulation of Tfh cells and the resultant germinal center responses by directly targeting ICOS[12]. Similarly, a potential involvement of miR-146a in controlling B cell responses has also been proposed[11,13]. Nevertheless, clear mechanistic insights into miR-146a-mediated B cell regulation are still lacking. More importantly whether miR-146a plays a non-redundant B cell-intrinsic role in maintaining optimal GC responses and preventing the development of autoimmunity remains to be further determined.

To directly examine the function of miR-146a in regulating B cell responses, we generated mice harboring a conditional allele of miR-146a, which allows for cell-type-specific miR-146a ablation. Our results demonstrate that miR-146a deletion in B cells alone is sufficient to lead to the development of spontaneous GC responses and the production of autoantibodies as mice aged. Upon immune challenges, exaggerated GC reactions observed in mice harboring miR-146a-deficient B cells is associated with increased production of high affinity antibodies. Mechanistically, miR-146a represses many CD40 signaling pathway components, which likely contribute to the heightened GC responses in mice with a B cell-specific miR-146a ablation. On the other hand, only when both miR-146a and its paralog, miR-146b, were deleted in T cells can increased Tfh cells be detected. Together, our data suggest that members of the miR-146 family serve as crucial cell-intrinsic molecular brakes to control B and T cells, two key arms in orchestrating protective humoral immunity. Coordinated regulations of GC reactions by miR-146a and miR-146b ensure the production of appropriate humoral immune responses without inducing undesired autoimmunity.

## Results

**miR-146a ablation in B cells led to spontaneous GC reactions.** Previously, a miRNA expression profiling of different mouse immune cell populations showed that among all T and B cell subsets, miR-146a is expressed at the highest level in Tfh and GC B cells, respectively[5]. Although miR-146a was recently suggested to restrain both Tfh and GC B cell responses[12], unlike what has been shown in Tfh cells, the precise molecular mechanism through which miR-146a controls GC B cell responses remains elusive. Moreover, considering that miR-146a is able to restrain a wide range of innate and adaptive immune cell populations, it is also unclear as to whether loss of miR-146a in B cells alone is sufficient to cause not only heightened GC responses but also autoimmunity. Thus, we first sought to validate the expression of miR-146a in GC B cells. Consistent with the previous report[5], we observed a clear increase in miR-146a expression in GC B cells when compared to naive mature B cells (Supplementary Fig. 1a). To directly assess the role of miR-146a in regulating GC B cell responses, we generated mice harboring a miR-146a allele that can be conditionally deleted in a desired cell population (Supplementary Fig. 1b). The resulting *miR-146a$^{fl}$* mice were bred to *CD21-cre* mice to induce mature B cell-specific miR-146a ablation (B KO) (Supplementary Fig. 1c)[14]. These mice appeared to be normal and did not exhibit the T cell activation phenotype that was observed in mice with the germ-line miR-146a deficiency around the same age (2–3 months), as described previously[11]. Nevertheless, a minimal increase in GL7$^+$PNA$^+$ GC B cells could already be detected in B-KO mice compared to WT littermates (Supplementary Fig. 1d). The elevated GC B cell phenotypes were further exaggerated when mice aged. As shown in Fig. 1a, b, at the age of 6 months a clear increase in GC B cell frequencies was readily detectable in B-KO mice along with elevated Tfh cell numbers. Moreover, immunohistochemistry showed that the spleens of aged B-KO mice contained considerably more and larger GCs with elevated numbers of infiltrating Tfh cells than did their WT littermates (Fig. 1c). Finally, these B-KO mice exhibited increased amounts anti-double-stranded DNA (dsDNA) autoantibodies similar to what has been reported in miR-146a germline deficient mice (Fig. 1d)[15]. Altogether, our findings suggest that miR-146a plays a critical role in maintaining optimal B cell responses and that loss of miR-146a in B cells is sufficient to lead to the development of spontaneous autoimmunity.

**Elevated humoral immune responses in B-KO mice.** Having documented the B-KO mice phenotype at steady state, we next sought to examine the humoral immune responses elicited upon antigenic challenge in these mice. To this end, we first immunized young B-KO mice prior to the development of spontaneous GC reactions with sheep red blood cells (SRBCs). As shown in Fig. 2a, b, B-KO mice harbored significantly increased frequencies of not only GC B cells but also Tfh cells, indicating the presence of an enhanced GC reaction. Supporting these findings, we also detected heightened GC responses with increased numbers of infiltrating Tfh cells in the spleens of immunized B-KO mice (Fig. 2c). Consequently, elevated plasma cell numbers as well as substantially increased production of SRBC-specific IgG was also observed in B-KO mice (Fig. 2d, e and

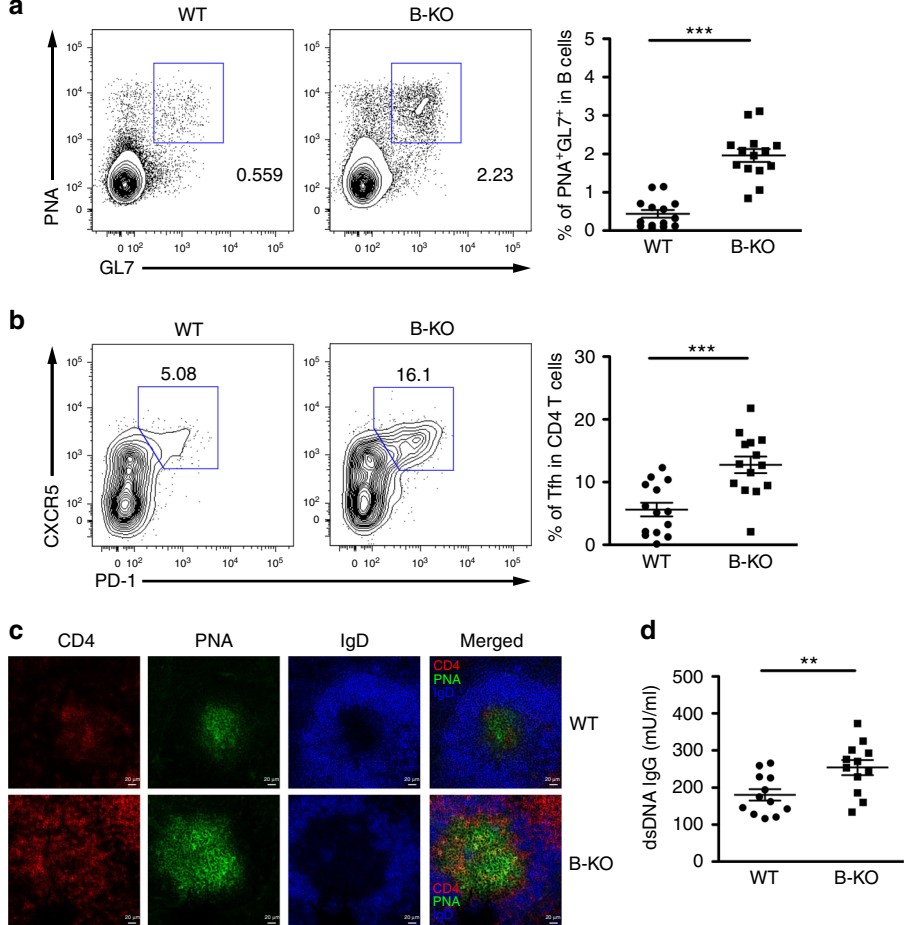

**Fig. 1** Loss of miR-146a in B cells led to spontaneous GC reactions in aged mice. FACS analyses and frequencies of **a** PNA$^+$GL7$^+$ GC B cells and **b** CXCR5$^+$PD-1$^+$ Tfh cells in spleen from ~6 months old B-KO mice or their WT littermates. **c** Immunohistological analyses of GC reactions in SRBC-immunized spleen that were cryocut and stained with CD4 (red), PNA (green), and IgD (blue). **d** ELISA analysis of serum anti-dsDNA autoantibodies in aged B-KO or control mice. Data are representative of three independent experiments. Each symbol represents an individual mouse, and the bar represents the mean. Significance was determined by unpaired Student's $t$-test with a 95% confidence interval. **$p < 0.01$, ***$p < 0.001$

Supplementary Fig. 2). On the other hand, when miR-146a ablation was restricted to GC B cells, only mild increases in both GC B cell and Tfh cell numbers could be detected at day 28 post SRBC immunization (Supplementary Fig. 3), suggesting a minimal role of sustained miR-146a expression in GC B cells in regulating GC reactions. To further examine the impact of B cell-specific miR-146a deficiency on humoral immunity, B-KO mice were then immunized with 4-hydroxy-3-nitrophenyl linked to Keyhole Limpet Hemocyanin (KLH) (NP-KLH) precipitated in alum. Immunization with NP-KLH allows for the detection of antibodies specific for the hapten NP, and the quantitative assessment of their affinity maturation[16]. As shown in Fig. 2f–h, while already exhibiting an increased, albeit not statistically significant, production of total NP-specific IgG1 antibody, B-KO mice produced significantly greater amounts of high-affinity NP-specific IgG1 antibody compared to their WT littermates. Altogether, these results demonstrated a cell-intrinsic role of miR-146a in B cells in controlling not only the magnitude but also the quality of humoral immunity likely through regulating the initial phase of the GC reaction.

**miR-146a targets multiple genes associated with CD40 pathway.** To gain further insight into molecular mechanisms that could account for miR-146a-mediated regulation of GC B cell responses, naive B cells from unimmunized B-KO mice or WT littermates and GC B cells from corresponding D14 SRBC-immunized mice were isolated and subjected to gene expression profiling analyses. Gene set enrichment analysis (GSEA) revealed substantial enrichment of genes related to immune cell activation, cytokine responses, as well as several different intracellular signaling pathways differentially expressed between WT and B-KO naive and/or GC B cells (Fig. 3a). Among them, miR-146a deficiency resulted in the most significant increases in genes involved in the NFκB signaling pathway in both naive and GC B cells (Fig. 3b). The role of the NFκB signaling pathway in controlling GC reactions has been well studied[17,18]. Particularly, activation of the NFκB signaling by CD40 engagement on B cells was shown to be critical in driving GC B cell differentiation[19]. Examination of the CD40 3′UTR revealed a potential miR-146a binding site, suggesting miR-146a might control GC B cell responses by directly targeting CD40 (Fig. 3c). However, even though CD40 protein levels were significantly higher in B cells devoid of miR-146a, our luciferase reporter results suggested that CD40 is not a miR-146a target (Fig. 3d, e). Nevertheless, it has been previously shown that CD40 stimulation can lead to further upregulation of CD40 itself as well as other molecules required for GC B cell differentiation[20]. Therefore, miR-146a might maintain optimal CD40 expression through direct regulation of components of the CD40 signaling pathway other than CD40. To this end, TRAF6 and Rel-B, two miR-146a targets which have been previously

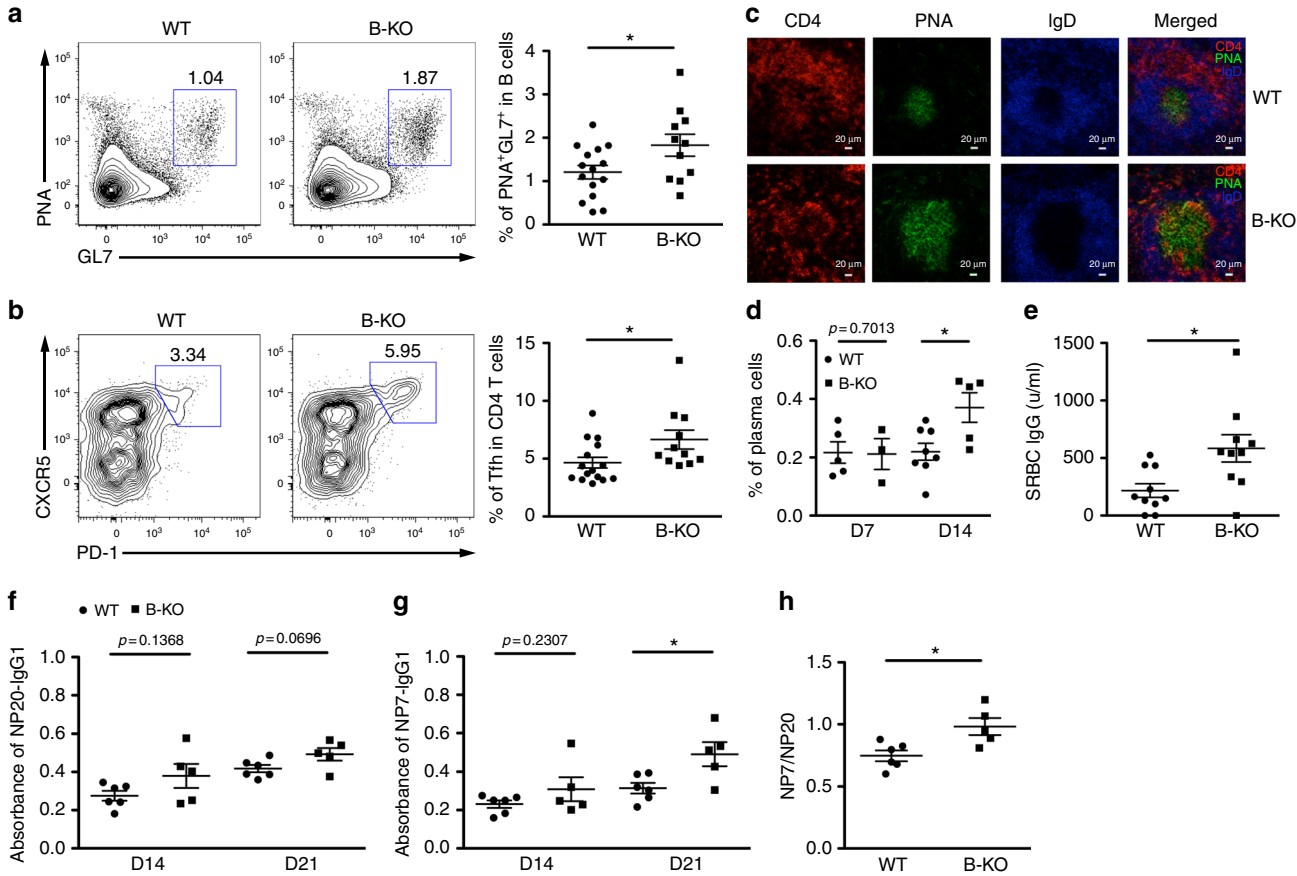

**Fig. 2** Elevated humoral immunue responses in mice with B cell-specific miR-146a ablation. FACS analyses and frequencies of **a** PNA+GL7+ GC B cells and **b** CXCR5+PD-1+ Tfh cells in spleen from ~8 wks old B-KO mice or their WT littermates 14 days after SRBC immunization. **c** Immunohistological analyses of GC reactions in SRBC-immunized spleen that were cryocut and stained with CD4 (red), PNA (green), and IgD (blue). Data are representative of four independent experiments. **d** Frequencies of CD138+Blimp1+ plasma cells in spleen from indicated mice at days 7 and 14 after SRBC immunization. ELISA analyses of **e** serum SRBC-specific IgG levels from SRBC-immunized B-KO mice or WT littermates or serum **f** total NP-specific IgG1 (NP20) and **g** high-affinity NP-specific IgG1 (NP7) responses from NP-KLH immunized B-KO mice and WT littermates at indicated time points. **h** The ratios of high-affinity and total IgG1 responses to NP antigen 21 days post immunization are shown. Data are representative of two independent experiments. Each symbol represents an individual mouse, and the bar represents the mean. Significance was determined by unpaired Student's *t*-test with a 95% confidence interval. *$p < 0.05$

characterized in other immune cell types[7,10,21], have both been shown to be key components in CD40-mediated NFκB activation in driving GC B cell responses[22,23]. Moreover, overexpression of Rel-B in B cells alone was also reported to be sufficient for upregulating CD40 expression[24]. Therefore, elevated levels of TRAF6 and Rel-B in B-KO B cells could likely contribute to our findings of enriched NFκB signaling gene signatures and enhanced CD40 expression (Supplementary Fig. 4a and b). Further examination of molecules downstream of the CD40 signaling pathway[25], suggested that *IKKα* and *c-Rel*, two genes known for their roles in GC B cell differentiation[26,27], may serve as additional miR-146a targets (Fig. 3f, g). Indeed, loss of miR-146a in B cells resulted in significantly increased amounts of both IKKα and c-Rel proteins, and our luciferase reporter studies confirmed that miR-146a is capable of direct repression of these two genes (Fig. 3h–k). Thus, miR-146a can target multiple molecules within the CD40 signaling pathway to ensure proper GC B cell responses. Consistent with this notion, we have found that many previously identified CD40 targets were upregulated in B cells with miR-146a deficiency (Fig. 3l)[28]. Moreover, miR-146a may also control CD40 expression without directly impacting the CD40 signaling pathway. STAT1, a known miR-146a target[9], has been shown to play an indispensable role in IFNγ-driven CD40 induction[29]. Therefore, elevated STAT1 levels in miR-146a-deficient B cells might also contribute to the enhanced CD40

expression phenotype (Supplementary Fig. 4c). Finally, consistent with the established role of miR-146a in functioning as a negative feedback regulator in controlling TLR and IFNγ signaling pathways[7,9], up-regulation of miR-146a expression was also observed in B cells in response to CD40 stimulation (Fig. 3m).

**Partial CD40 deletion rescued GC phenotypes in B-KO mice.** We observed that the loss of miR-146a in B cells resulted in not only enhanced CD40 signaling but also increased CD40 expression, suggesting that exaggerated CD40-dependent gene expression is likely responsible for the overexuberant humoral immunity. Moreover, considering that CD40 signaling was previously shown to act predominantly in the initial phase of the GC reaction[20], the aforementioned findings in mice with GC B cell-specific miR-146a ablation further support the notion that miR-146a controls GC B cell responses mainly through regulating the CD40 signaling pathway. Nevertheless, the miR-146a targets identified in our study are also involved in non-CD40 signaling pathways. Thus, we sought to determine whether restoring CD40 expression alone in B-KO B cells is sufficient in rescuing the abnormal GC phenotypes resulting from miR-146a deficiency. As shown in Fig. 4a, the amounts of CD40 in miR-146a-deficient B cells could be substantially reduced to the levels close to those in WT cells by removing one allele of *CD40* in B-KO mice (B-KO/

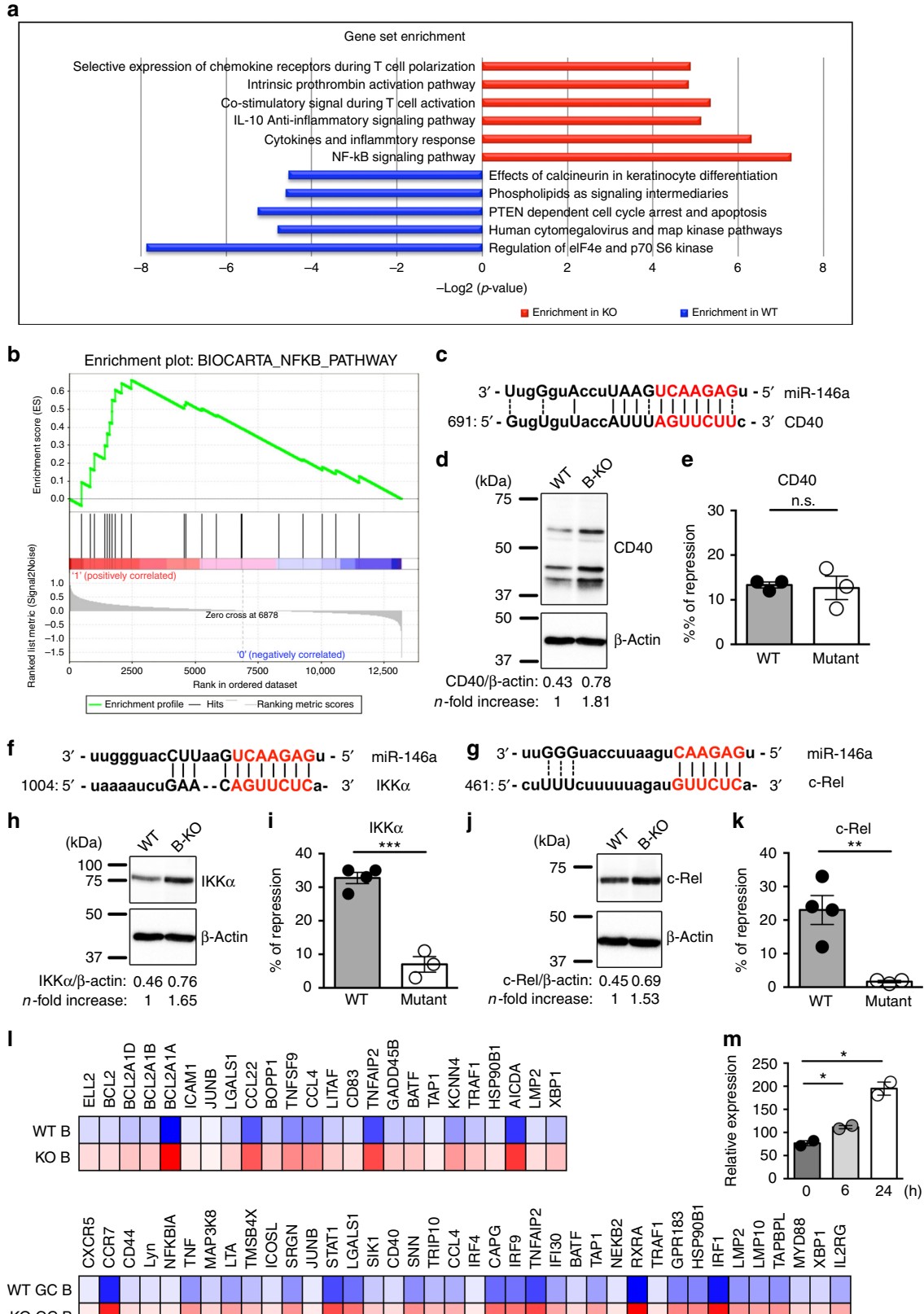

CD40[+/−]). However, unlike B cell-specific miR-146a ablation in B-KO mice, partial CD40 deletion was not restricted to B cells. Therefore, to examine the cell-intrinsic role of miR-146a-mediated regulation of CD40 expression in controlling GC B cell responses, naive B cells isolated from B-KO/CD40[+/−] mice

were first co-cultured with 3T3 feeder cells expressing CD40L and BAFF in the presence of IL-4 to induce GC phenotypes in vitro[30]. Consistent with our observations in B-KO mice, upon in vitro stimulation, miR-146a-deficient B cells exhibited significantly increased levels of activation-induced cytidine deaminase (AID),

**Fig. 3** Multiple genes in the CD40-NFκB axis are targeted by miR-146a in B cells. **a** GSEA analysis of the miR-146a deficiency gene signatures. Annotated GO biological processes assigned to genes differentially expressed in naive and/or GC B cells isolated from B-KO mice or WT littermates, as determined by RNA-seq. **b** GESA results of the NFκB pathway derived from the BioCarta pathway database were shown and a false discovery rate (FDR) of less than 0.05 was accepted as highly significant. Sequence alignment of the putative miR-146a binding site in the 3′ UTR of **c** CD40, **f** IKKα or **g** c-Rel. Expression of **d** CD40, **h** IKKα or **j** c-Rel in B-KO B cells vs. WT B cells were assessed by immunoblotting. Densitometric values of CD40, IKKα or c-Rel were normalized to β-actin expression values and n-fold increase on the basis of each corresponding WT. Ratios of repressed luciferase activity of cells in the presence of WT or mutated **e** CD40, **i** IKKα or **k** c-Rel 3′ UTR transfected with miR-146a compared with cells transfected with control empty vector. **l** Heatmap of representative CD40–targeted genes differentially expressed in naive or GC B cells in the presence or absence of miR-146a. **m** Quantitative PCR of miR-146a levels in B cells upon CD40 stimulation in vitro at different time points. Data represent the mean ± SD and are representative of 3 independent experiments ($n = 3$–6). Significance was determined by unpaired Student's $t$-test with a 95% confidence interval. *$p < 0.05$, **$p < 0.01$, ***$p < 0.001$

a CD40-induced enzyme that is crucial for GC B cell differentiation and function (Fig. 4b)[31]. On the other hand, the AID expression was greatly reduced when miR-146a deficiency was coupled with *CD40* hemizygosity. Similarly, the heightened levels of IgG1 expression observed in B-KO B cells was also restored to WT levels in B cells isolated from B-KO/CD40$^{+/-}$ mice (Fig. 4c). Finally, to test this in an in vivo setting, we performed a series of mixed bone marrow (BM) chimera studies similar to those described previously (Fig. 4d)[9]. In brief, the Ly5.1$^-$ GC B cell frequency in each chimeric mouse was normalized against the corresponding WT Ly5.1$^+$ GC B cell counterpart to account for the variation in the reconstitution and immunization efficiency in the mixed BM chimera study. As shown in Fig. 4e, f and Supplementary Fig. 5, by comparing GC B cells in the B-KO/CD40$^{+/-}$ compartment of chimeric mice to those in the B-KO or WT counterparts at various time points post SRBC immunization, we have found that the elevated GC B cell responses observed in mice harboring miR-146a-deficient B cells were largely alleviated upon partial CD40 deletion. On the other hand, no alteration could be detected in total B cell frequencies between the WT, B-KO and B-KO/CD40$^{+/-}$ groups during the course of SRBC immunization (Supplementary Fig. 6). Taken together, these results clearly pointed to the CD40 signaling pathway as the major target of miR-146a in B cells to regulate GC B cell responses and the resultant humoral immunity.

**No further enhanced humoral responses in TB-KO mice.** Our experiments demonstrated that expression of miR-146a in B cells is necessary for maintaining optimal GC B cell responses. Given that miR-146a was also shown to limit the accumulation of Tfh cells[12], one would expect that miR-146a controls GC reactions through limiting GC B and Tfh cells concurrently and that deletion of miR-146a in B and T cells together may lead to further escalation of humoral responses. To test this hypothesis, *miR-146a$^{fl}$* mice were bred to hCD2-cre mice to generate a mouse line with miR-146a ablation in B and T cells (TB-KO) (Supplementary Fig. 7)[32]. Surprisingly, even though elevated GC B cell frequencies were readily detectable in TB-KO mice upon SRBC immunization (Fig. 5a), there was no further increase in GC B cell numbers in TB-KO mice compared to those observed in B-KO mice (Fig. 5b). Likewise, despite having miR-146a ablated in both B and T cells, the increase in Tfh cell frequencies was also similar between TB-KO and B-KO mice (Fig. 5c, d). Consistently, despite the fact that TB-KO mice displayed augmented SRBC-specific antibody responses (Fig. 5e), additional loss of miR-146a in T cells did not result in further enhanced humoral responses compared to mice exhibiting miR-146a deficiency only in B cells (Fig. 5f).

**Negligible impact from T cell-specific miR-146a ablation.** The aforementioned results obtained from TB-KO mice raised a question as to whether miR-146a is required to directly control Tfh cell responses. Previously, the cell-intrinsic role of miR-146a in limiting Tfh cell numbers was suggested based on studies

utilizing either adoptive T cell or mixed BM stem cell transfers[12]. As those approaches involve the transfer of cells into mice with Treg cell defects (i.e., CD28$^{-/-}$) or lymphodeplete recipients (i.e., Rag1$^{-/-}$) in which biological manifestations could be potentially amplified[9,33], we next sought to determine whether miR-146a deficiency limited to T cells alone can result in augmented Tfh cell as well as overall GC responses in lymphoreplete mice. To this end, we generated another mouse line in which miR-146a is specifically deleted in T cells by crossing *miR-146a$^{fl}$* mice to *CD4-cre* mice (T-KO) (Supplementary Fig. 8a). miR-146a was previously shown to be involved in restraining IFNγ-mediated Th1 immunity through targeting Stat1 in both Th1 and Treg cells. Therefore, to prevent the potential confounding effects of pre-existing Th1 responses in T-KO mice, Tfh cells were assessed in mice at 2–3 months of age prior to the onset of unprovoked IFNγ responses. As shown in Supplementary Fig. 8b and c, at steady state, no alteration in the frequencies of Tfh cells as well as GC B cells in T-KO mice could be detected. Moreover, SRBC-immunized T-KO mice harbored equivalent numbers of Tfh and GC B cells compared to their WT littermates (Fig. 6), indicating that loss of miR-146a in T cells alone had minimal impact compared to loss of miR-146a in B cells.

**miR-146a and miR-146b jointly regulate Tfh cell responses.** There are two members of the miR-146 family. Whereas the *miR-146a* gene is located on mouse chromosome 11, its paralog, *miR-146b*, is located on chromosome 19 (chromosomes 5 and 10 in human, respectively). As their mature sequences differ by only two nucleotides and both have the same seed sequence, miR-146a and miR-146b likely regulate the same set of targets[34]. However, all functional studies thus far have focused solely on miR-146a, likely due to the fact that in most immune cell subsets miR-146a is constitutively highly expressed or upregulated in immune responses[6]. Consistent with a previous report[5], we observed lower levels of miR-146b, compared to miR-146a, in both Tfh and GC B cells (Supplementary Fig. 9a). Nevertheless, despite its low abundance, expression of miR-146b in T cells might still enables normal control of Tfh cell responses in the absence of miR-146a. To directly test this possibility, we generated mice harboring a conditional miR-146b allele (*miR-146b$^{fl}$*) (Supplementary Fig. 9b). Similar to what was observed in mice with T cell-specific deletion of miR-146a, loss of miR-146b in T cells alone did not result in any detectable phenotype (Supplementary Fig. 10). However, ablation of both miR-146a and miR-146b in T cells (T-DKO) (Supplementary Fig. 9c), resulted in spontaneous accumulation of Tfh and GC B cells even in mice at a young age (2~3 months) (Fig. 7a, b). Upon SRBC immunization, T-DKO mice exhibited further increases in numbers of Tfh and GC B cells compared to their WT littermates (Fig. 7c, d). Consequently, elevated levels of SRBC-specific IgG could be readily detected in T-DKO mice, while T-KO and WT mice exhibited comparable SRBC-specific antibody responses (Fig. 7e). Finally, we sought to determine whether miR-146b contributes to the regulation of

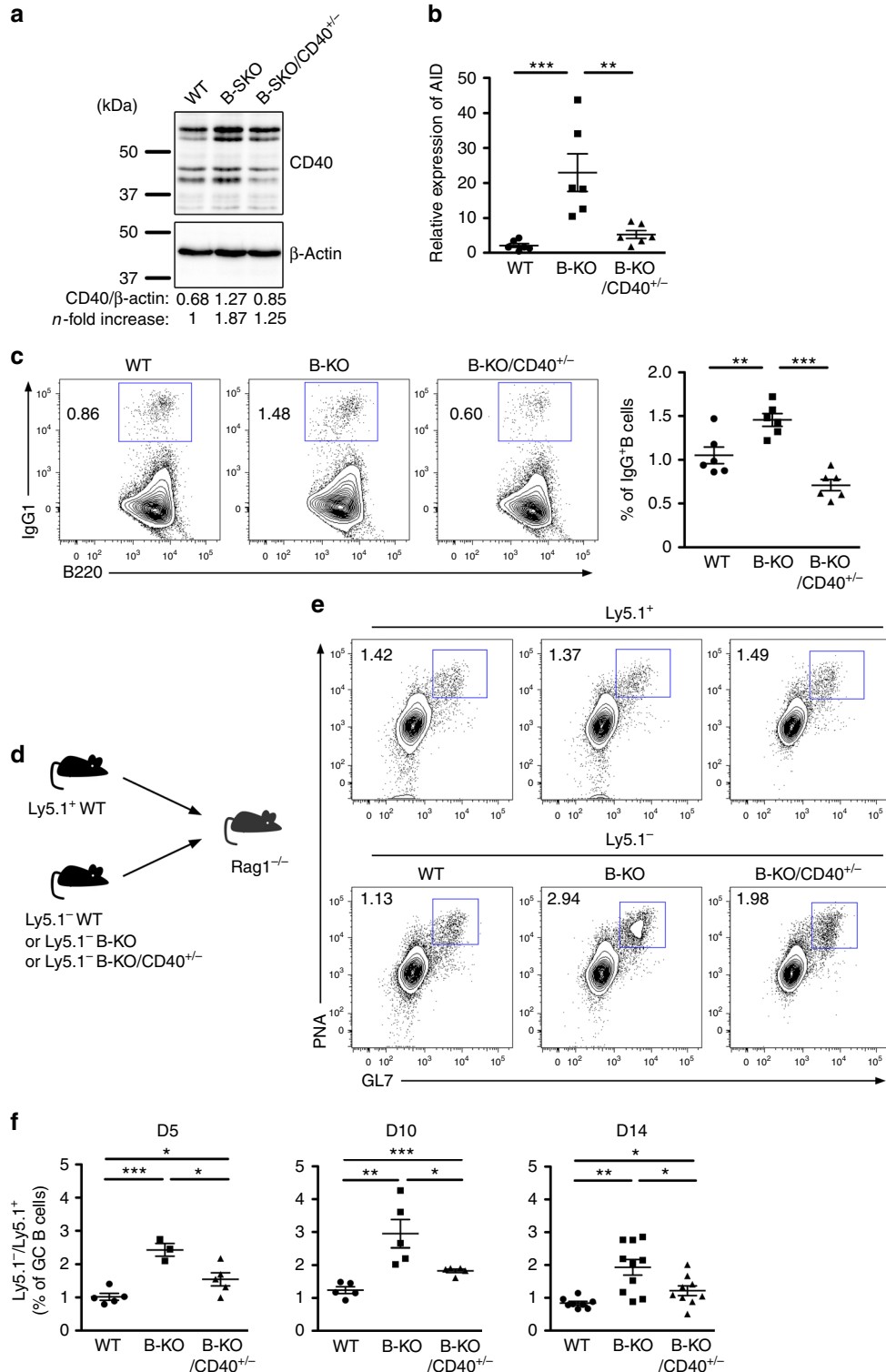

**Fig. 4** Partial CD40 deletion was able to largely rescue the aberrant GC B phenotypes resulting from miR-146a deficiency. **a** Expression of CD40 in B-KO B cells, B-KO/CD40$^{+/-}$ B cells *vs.* WT B cells was assessed by immunoblotting. Densitometric values of CD40 were normalized to β-actin expression values and n-fold increase on the basis of WT controls. **b** Quantitative PCR of AID mRNA levels and **c** FACS analysis and frequencies of IgG1$^+$ cells in in vitro CD40-stimulated B cells from B-KO, B-KO/CD40$^{+/-}$ or WT mice are shown. **d** Schematic of generation of mixed BM chimeras. **e** FACS analyses of Ly5.1$^-$ and Ly5.1$^+$ PNA$^+$GL7$^+$ GC B cells in spleen from indicated mixed BM chimeras 14 days after SRBC immunization. **f** Ratios of frequencies of Ly5.1$^-$ and Ly5.1$^+$ GC B cells in spleen from indicated chimeric mice at days 5, 10 and 14 days after SRBC immunization. Data are representative of three independent experiments. Each symbol represents an individual mouse, and the bar represents the mean. Significance was determined by unpaired Student's *t*-test with a 95% confidence interval. *$p < 0.05$, **$p < 0.01$, ***$p < 0.001$

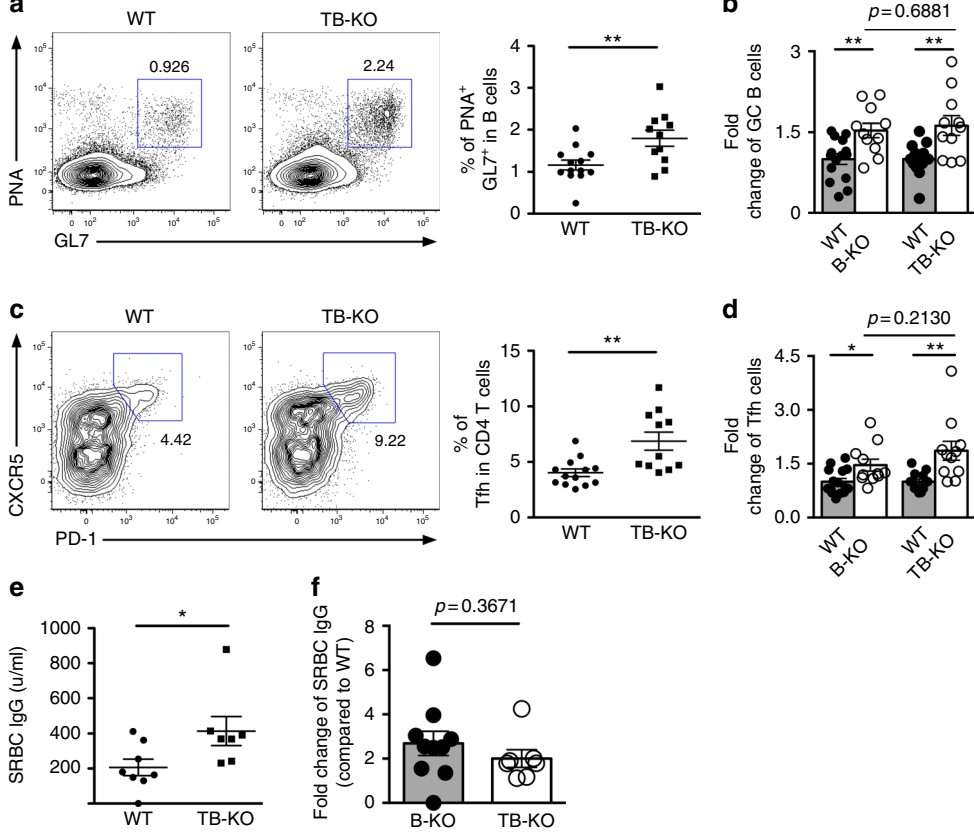

**Fig. 5** Additional miR-146a ablation in T cells did not result in further enhanced humoral immune responses in mice harboring B cells devoid of miR-146a. FACS analysis and frequencies of **a** PNA+GL7+ GC B cells or **c** CXCR5+PD-1+ Tfh cells from spleen of ~8 wks old TB-KO mice or their WT littermates 14 days after SRBC immunization. Fold changes (on the basis of corresponding WT controls) of **b** GC B or **d** Tfh cell frequencies in B-KO and TB-KO mice are shown. **e** ELISA analyses of serum SRBC-specific IgG levels in immunized TB-KO mice or WT littermates. **f** Fold changes (on the basis of corresponding WT controls) of serum SRBC-specific IgG levels from B-KO and TB-KO mice. Data represent the mean ± SD and are representative of three independent experiments. Each symbol represents an individual mouse, and the bar represents the mean. Significance was determined by unpaired Student's t-test with a 95% confidence interval. *$p < 0.05$, **$p < 0.01$

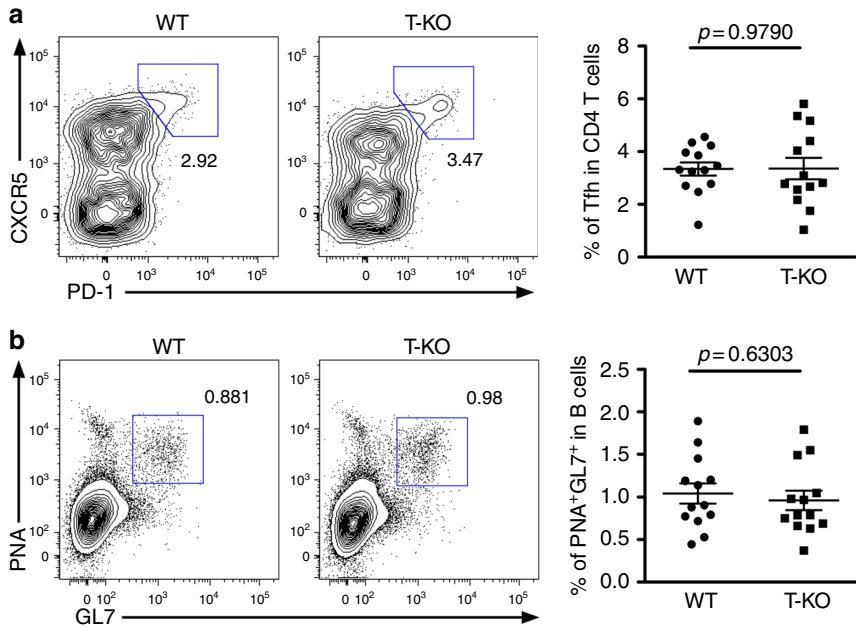

**Fig. 6** Loss of miR-146a in T cells alone is not sufficient to impact Tfh cells and the corresponding GC B cells. FACS analyses and frequencies of **a** CXCR5+PD-1+ Tfh cells or **b** PNA+GL7+ GC B cells from spleen of ~8 wks old T-KO mice or their WT littermates 14 days after SRBC immunization. Data represent are representative of 4 independent experiments. Each symbol represents an individual mouse, and the bar represents the mean

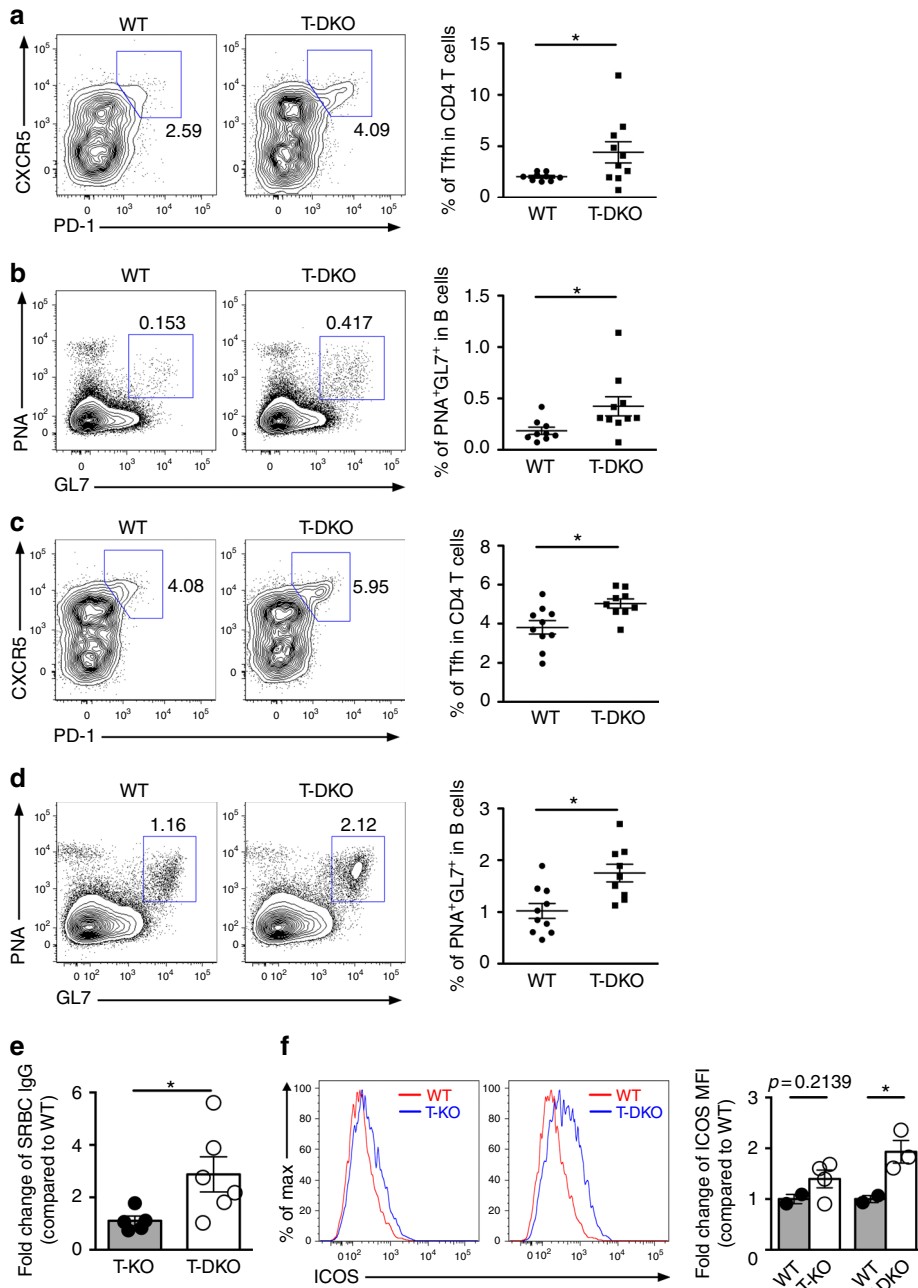

**Fig. 7** Both members of miR-146 family are redundantly essential in maintaining normal Tfh cell responses. FACS analyses and frequencies of **a** CXCR5 $^+$PD-1$^+$ Tfh cells or **b** PNA$^+$GL7$^+$ GC B cells from spleens of ~8 wks old T-DKO mice or their WT littermates. FACS analyses and frequencies of **c** CXCR5 $^+$PD-1$^+$ Tfh cells or **d** PNA$^+$GL7$^+$ GC B cells from spleens of ~8 wks old T-DKO mice or their WT littermates 14 days after SRBC immunization. **e** Fold changes (on the basis of corresponding WT controls) of serum SRBC-specific IgG levels from T-KO and T-DKO mice. **f** FACS analyses of the expression of ICOS in CD44$^{hi}$CD4$^+$ T cells in spleen from T-KO and T-DKO mice compared to their corresponding WT littermates. Fold changes (on the basis of corresponding WT controls) of ICOS mean fluorescence intensity (MFI) were shown in the right panels. Data represent the mean ± SD and are representative of three independent experiments. Each symbol represents an individual mouse, and the bar represents the mean. Significance was determined by unpaired Student's *t*-test with a 95% confidence interval. *$p < 0.05$

genes that are known to be targeted by miR-146a. As noted above, miR-146a was suggested to limit Tfh cell numbers through repressing ICOS[12]. While ICOS expression was seemly increased in the absence of miR-146a consistent with the aforementioned report, significantly enhanced ICOS expression could only be detected when miR-146a and miR-146b were simultaneously ablated (Fig. 7f). Altogether, our data show that both members of miR-146 family are redundantly required to control Tfh cells, likely through coordinately targeting the same set of molecules.

## Discussion

The GC reaction, a hallmark of B cell-mediated immune responses to T cell-dependent (TD) antigens, is critical for the generation of long-term humoral protection. Positive feedback signals between B cells and Tfh cells through the CD40–CD40L and ICOSL–ICOS interactions support the formation of GCs[35]. Moreover, the strong correlation between GC B and Tfh cell numbers further highlights a tight regulation between two major cellular players in GCs[36]. Previously, miR-146a was shown to be

highly induced in both GC B and Tfh cells[5]. It was suggested that elevated expressions of miR-146a in Tfh cells is crucial for maintaining optimal Tfh cell numbers through targeting ICOS[12]. In our work, by using cell-type-specific targeting approaches, we have shown that miR-146a controls GC reactions primarily through regulating the CD40 signaling pathway in GC B cells and that loss of miR-146a in B cells is sufficient to lead to the development of spontaneous autoimmunity. In addition, our data identified a previously unrecognized role of miR-146b in collaborating with miR-146a to regulate Tfh cell responses and the resultant humoral immunity.

Interestingly, while loss of miR-146a-mediated regulation resulted in CD40 upregulation, we did not detect a clear difference in surface CD40 expression between WT and B-KO B cells. Previously, it has been shown that CD40 signaling induces rapid endocytosis of CD40 and internalization of CD40 is a way to negatively regulate its signaling[37,38]. Therefore, enhanced CD40 signaling in B-KO B cells would likely lead to increased CD40 internalization, further supporting the notion that miR-146a functions as a key negative regulator in controlling CD40 signaling. Nevertheless, it should be noted that many known or new targets of miR-146a identified in the past or in the present study are not restricted to the CD40 pathway alone. In fact, miR-146a was first documented to regulate TLR signaling through targeting IRAK1 and TRAF6[7]. Whereas TLR stimulation was initially thought to be mainly involved in driving the production of extrafollicular antibodies, recent studies have shown that TLRs can also promote GC B cell responses to produce high affinity class-switched antibodies[39]. Moreover, like CD40, another tumor necrosis factor receptor (TNFR) superfamily member, BAFF receptor, also transduces its downstream signals in B cells through engaging TRAF6 and NFκB and was suggested to be critical for GC generation[40,41]. Finally, in addition to its aforementioned role in inducing CD40 expression, IFNγ-driven STAT1 activation in B cells was also shown to promote spontaneous GC formation in a T-bet-dependent manner[42]. Therefore, even though our data has clearly demonstrated that miR-146a-mediated regulation of CD40 signaling plays a major role in controlling GC B cell responses, it is conceivable that dysregulation of the aforementioned signaling pathways in the absence of miR-146a-mediated control in B cells could contribute to the observed enhanced GC responses.

To date, increasing evidence has pointed to miR-146a as a major molecular regulator in the immune system[6]. In addition to its role in negatively regulating many different innate and adaptive immune cell populations[10,43], miR-146a was also shown to be required for proper Treg cell-mediated immune regulation[9]. As such, it is difficult to pinpoint the precise role of miR-146a in controlling humoral immunity, as the spontaneous accumulation of Tfh and GC B cells observed in miR-146a germ-line null mice might either reflect cell-intrinsic defects or result from functional abnormalities in other cell types. To this end, by employing the conditional gene targeting strategy, our study has clearly demonstrated that miR-146a deficiency limited to B cells alone is sufficient to impact the overall GC responses. On the other hand, unlike what was reported previously[12], miR-146a does not seem to play an essential role in controlling Tfh cells, as deletion of miR-146a in T cells did not lead to any changes in Tfh cell numbers nor resulted in further enhanced humoral responses when combined with miR-146a deficiency in B cells. While this finding was initially surprising, it is not entirely unexpected. The difference in the experimental systems used between these two studies is likely to be accounted for the observed discrepancy. Impaired Treg cell-mediated regulation or the transient lymphopenia that resulted from the adoptive cell transfer or mixed BM chimera approaches taken in the former study could both

potentially cause accelerated and/or enhanced phenotypes which might not be revealed in unperturbed mice with cell-specific gene deletion[9,44]. Therefore, while the expression of miR-146a in T cells may still be required under certain extreme conditions, it is dispensable for maintaining optimal Tfh cell responses in regular settings,

Our studies revealed a previously unappreciated role of miR-146b in Tfh cells. Despite its relatively low expression levels, miR-146b was capable of controlling Tfh cell responses in the absence of miR-146a. On the other hand, expression of miR-146b is insufficient to compensate for the loss of miR-146a in B cells, which is likely due to the fact that B cells, when compared to T cells, express even lower amounts of miR-146b. Nevertheless, it is possible that deletion of both miR-146a and miR-146b in B cells could further enhance GC B cell responses. Mechanistically, our data suggest that miR-146b could exert its regulatory function in Tfh cells through targeting the same set of genes (e.g., ICOS) that are repressed by miR-146a, as they share the same seed sequence. As such, only when both miR-146 members are simultaneously deleted, the changes in gene expression levels would be large enough to impact the Tfh cell biology. Moreover, considering that these two miRNA paralogs do differ by two nucleotides in their mature sequences, it is also possible that they could have unique regulatory activities determined by sequence characteristics outside of the seed region[45]. Thus, miR-146b could also coordinately regulate Tfh cell responses with miR-146a through targeting different molecules that are in a common biological pathway or shared protein complex.

Considering the dominant role of miR-146 in controlling both innate and adaptive immunity, it is not surprising that abnormal expression of miR-146a (and to a lesser degree, miR-146b) has been associated with many different pathological conditions[45,46]. Specifically, in systemic lupus erythematosus (SLE) patients, the expression level of miR-146a is negatively correlated with the disease pathogenesis[47,48]. Dysregulated type I IFN signaling resulted from diminished miR-146a expression was thought to be responsible for the increased SLE susceptibility in the aforementioned studies. However, previously it has also been shown that many SLE disease activities in patients such as proteinuria, B cell hyperactivation and autoantibody production could all be greatly ameliorated by CD40 blockade[49]. Therefore, it is tempting to speculate that loss of miR-146a-mediated CD40 regulation in B cells could also play an important pathogenic role in contributing to this complex autoimmune disorder.

In conclusion, our study delineates cellular and molecular mechanisms underlying regulation of GC responses by miR-146. Analogous to Bcl6, the two miR-146 family members coordinately regulate GC B and Tfh cells, two key arms in orchestrating protective humoral immunity. These findings will undoubtedly facilitate the development of successful strategies aimed at targeting miR-146-mediated gene regulatory machineries in both GC B and Tfh cells to generate robust and long-lasting immune responses against a wide array of invading microorganisms as well as to prevent unwanted autoimmunity.

## Methods

**Mice.** CD21-cre[14], Cγ1-cre[50], CD4-cre[51], hCD2-cre[32], and CD40-deficient mice[52] were purchased from the Jackson Laboratory. The targeting constructs for miR-146a[fl53] and miR-146b[fl] were generated using recombineering. For detailed information see the National Cancer Institute recombineering website (http://redrecombineering.ncifcrf.gov/). Mice on a B6 genetic background with the germ-line insertion were bred to Flp deleter mice to remove the neomycin resistance cassette. B cell-, T cell-, or T and B cell- specific deletion of miR-146a and/or miR-146b were achieved by breeding miR-146a[fl/fl] or miR-146b[fl/fl] mice to CD21-cre, CD4-cre or hCD2-cre mice respectively. CD40-deficient mice were bred with B-KO mice to generate B-KO/CD40[+/−] mice. Unless otherwise indicated, 8- to 10-weeks-old mice were used. All mice were maintained and handled in accordance with the Institutional Animal Care and Use Guidelines of UCSD and National Institutes of

Health Guidelines for the Care and Use of Laboratory Animals and the ARRIVE Guidelines.

**Immunization**. Mice were immunized intraperitoneally with $2 \times 10^9$ SRBCs (Colorado serum company) in PBS. Spleen and peripheral blood were harvested at the indicated time after SRBC immunization. For NP-KLH immunization, 100 μg of NP-KLH (Biosearch Technologies) in PBS mixed with 50% of alum (Thermo Scientific) were injected intraperitoneally, and then peripheral blood was collected on days14 and 21 after immunization.

**Bone marrow chimeras**. Bone marrow (BM) cells were isolated from WT, B-KO, B-KO/CD40$^{+/−}$, and Ly5.1$^+$ B6 mice, followed by T cell depletion using a mouse Pan T (90.2) kit (Dynabead). BM cells from WT, B-KO, and B-KO/CD40$^{+/−}$ mice were mixed with BM cells from Ly5.1$^+$ B6 mice at a 1:1 ratio, respectively, and transferred into lethally irradiated Rag1$^{−/−}$ recipients intravenously. After 8 weeks, recipients were immunized with SRBCs. Next, 14 days post immunization, spleens were harvested for FACS analysis.

**Flow cytometry and antibodies**. For FACS analysis, single-cell suspensions of spleens were stained with Ghost Dye Red 780 (Tonbo Biosciences 13-0865, 1:1000) or Fixable Viability Dye eflour 450 (Thermo Fisher Scientific 65-0863-14, 1:1000), followed by surface staining with antibodies against CD4 (Thermo Fisher Scientific 11-0041-82 or 45-0042-82, 1:500), CD8 (Thermo Fisher Scientific 47-0081-82, 1:500 or 48-0081-82, 1:1000, Biolegend 100747, 1:400), CD44 (Thermo Fisher Scientific 48-0441-80, 1:600), PD-1 (Thermo Fisher Scientific 25-9958-82, 1;400), B220 (Thermo Fisher Scientific 45-0452-82 or 25-0452-82, 1:500), CD3e (Thermo Fisher Scientific 47-0031-82, 1:250), CD138 (Biolegend 142503, 1:100), CD95 (BD Pharmingen 740716, 1;400), GL7 (Thermo Fisher Scientific 49-5902-82, 1:500), CD45.1 (Thermo Fisher Scientific 17-0453-82, 1:500), ICOS (Thermo Fisher Scientific 17-9949-82, 1:500), Thy1.1 (Thermo Fisher Scientific 12-0900-83, 1:1000), and PNA (Vector Laboratories, FL-1071, 1:800). For CXCR5, cells were stained with purified CXCR5 (BD Bioscience 551961, 1:100) for 1 h, followed by staining with biotinylated anti-rat IgG (Jackson 112-065-003, 1:1000) for 30 min, and then surface staining was performed with indicated antibodies, as well as PE- or APC-labeled streptavidin (Thermo Fisher Scientific 12-4317-82 or 17-4317-82, 1:1000). Intracellular staining was performed with antibodies against FOXP3 (Thermo Fisher Scientific 53-5773-82, 1:300), Blimp1 (Biolegend 150003, 1:100), and IgG1 (BD Bioscience 560089, 1:500). Data were collected by BD LSRFortessa or BD LSRFortessa X-20 (BD Biosciences) and analyzed by FlowJo (Tree star). FACS gating strategies are included in Supplementary Figs. 11 and 12.

**In vitro B cell stimulation**. Single-cell suspensions of spleen cells were stained with PE-CD19 (Biolegend 115508, 1:500), and then CD19$^+$ B cells were purified using anti-PE Microbeads (Miltenyl). In total $5 \times 10^9$ B cells were plated in a 48-well plate and cultured for 4 days in the presence of αCD40 (1 μg/ml, BioXcell BE-0016-2) and IL-4 (20 ng/ml, Peprotech 214-14), followed by standard staining for FACS analysis, as described above. To obtain GC B-like cells in vitro, purified B cells were co-cultured with CD40LB cells, as described previously[30]. Briefly, CD40LB cells were treated with mitomycin C (30 μg/ml, Sigma) at 37 °C for 30 min, and then $2.5–5 \times 10^4$ cells were plated in a 6-well plate. After 24 h, the media was removed, and then $5 \times 10^9$ B cells suspended in IL-4 (10 ng/ml)-containing media were added onto the CD40LB cells.

**Quantitative PCR analysis**. Cells were sorted on a FACSAria Fusion (BD Biosciences) and then total mRNA was isolated using a miRNEasy kit (Qiagen) according to the manufacturer's instructions. To determine expression of miR-146a and miR-146b, TaqMan MicroRNA Assay (Thermo Fisher Scientific) was performed. Real-time reactions were run on a 7900HT Fast Real-Time PCR System (Thermo Fisher Scientific). For in vitro stimulated B cells, after 3 days, PNA$^+$GL7$^+$B220$^+$ cells were FACS sorted. Total mRNA was isolated using a miRNEasy kit, and then cDNA was generated using iScript cDNA Synthesis Kit (Bio-Rad), followed by qPCR reactions using SYBR Green PCR Mix (Thermo Fisher Scientific). The primer sequences used were as follows: AID Forward 5′-CAAGCTTCCTTTGGCCTAAGAC-3′, AID reverse 5′- CAGCGGACATTT TTGAAATGG- 3′, GAPDH Forward 5′-CGTCCCGTAGACAAAATGGT-3′, GAPDH reverse 5′-TCAA TGAAGGGGTCGTTGAT-3′.

**Gene expression profiling analysis**. Spleens were harvested from unimmunized WT and B-KO mice, and then B220$^+$ cells were FACS sorted. GC B cells (B220$^+$PNA$^+$GL7$^+$) in spleens were sorted from SRBC-immunized WT and B-KO mice on D14. Poly-A RNA-seq was performed using three biological replicates for each cell population as described previously. RNA libraries were prepared for sequencing using standard Illumina library preparation protocols. Sequenced reads were trimmed for adaptor sequence, and masked for low-complexity or low-quality sequence. STAR was used to align reads to the mm10 whole genome[54]. RNA-seq experiments were normalized and the Reads per kilobase of transcript per megabase of library size (RPKM) values were generating for RefSeq annotated

transcripts using HOMER[55]. To analyze RNA-seq results, hierarchical clustering was performed (uncentered correlation distance and average linkage) using the average of the RPKM values of the triplicate biological experimental data sets to group the genes into similar regulatory circuits. The low expressed and invariant genes for which none of the datasets had more than 32 RPKM, or the highest and lowest expressions were less than a two-fold difference, were removed. All measurements were relative to the mean of the gene to generate a heatmap visualization of the results. Then, Gene Set Enrichment Analysis (GSEA) between KO versus WT, or KO GC B versus WT GC B datasets were applied to determine gene that are differentially expressed in naive and/or GC B cells between WT and B-KO mice, and to connect the expression data to the pathways with default parameters[56,57]. Gene sets derived from the BioCarta pathway databases were chosen and a false discovery rate (FDR) of <0.05 was accepted as highly significant.

**Immunoblotting**. Spleens were harvested from 6 to 8-weeks-old mice. Total CD19$^+$ B cells were MACS isolated with >95% purity (Miltenyi Biotech) and were subjected to lysis in RIPA buffer (Cell Signaling) supplemented with 1 mM PMSF. Next, cell lysates were separated by SDS-PAGE and transferred onto PVDF membranes. Antibodies against CD40 (Santa Cruz sc-974, 1:1000), TRAF6 (Santa Cruz sc-7221, 1:500), RelB (Santa Cruz, sc-226, 1:1000), c-Rel (Thermo Fisher Scientific 14-6111-82, 1:1000), IKKα (Cell signaling 2682, 1:1000), STAT1 (Cell signaling 9172, 1:1000), and Actin (Sigma A1978, 1:3000) were used to detect corresponding proteins. Protein levels were quantified by Image J (National Institues of Health). Uncropped immunoblot images are included in Supplementary Figs. 13 and 14.

**ELISA**. To assess autoantibody against dsDNA, serum was harvested from aged mice (~6 months) and mouse anti-dsDNA ELISA Kits (Shibayagi) were used to perform ELISA according to the manufacturer's instructions. To measure SRBC-specific IgG production, serum was harvested from mice on D14 after SRBC immunization and assessed by using SRBC IgG ELISA kits (Abnova) according to the manufacturer's instructions. For NP immunization experiments, mice were immunized with NP-KLH and NP-specific antibodies that bound to NP$_{20}$-BSA or NP$_7$-BSA (Biosearch Technologies) to be determined by ELISA. In brief, 96-well plates were coated overnight at 4 °C with NP$_{20}$-BSA or NP$_7$-BSA, followed by blockade of non-specific binding by incubation with blocking buffer (1% of BSA in PBS) for 1 h at room temperature. Mouse serum was diluted to $10^{-5}$ of the original concentration in blocking buffer and then added to the plates, followed by incubation for 1 h at room temperature. Plates were washed with washing buffer (0.05% Tween-20 in PBS) three times. Bound antibodies were detected by HRP-conjugated anti-mouse IgG1 antibodies (Southern Biotech 5300-05, 1:1000). The reactions were developed by incubation for 15 min at room temperature with TMB substrate (Biolegend) and were stopped by the addition of 2 N $H_2SO_4$. Absorbance was measured by a microreader (Bio-Rad) at 450 nm.

**Luciferase assay**. The 3′UTR sequences of CD40, IKK and c-Rel were amplified from mouse genomic DNA and cloned into psicheck2. The TRAF6 UTR sequence was synthesized (Integrated DNA Technologies) and cloned into psicheck2. Site-directed mutagenesis (Agilent) was performed to obtain 3′UTR mutants of CD40, IKK, c-Rel and TRAF6. Indicated 3′UTR WT or mutant plasmids were transfected into HEK293T (ATCC CRL-3216) cells along with either a miR-146a-expressing plasmid or a control empty vector. Luciferase activity was determined by the Dual luciferase reporter assay system (Promega) according to the manufacturer's instructions at 20 h after transfection.

**Histology**. Spleens were frozen in Tissue-Tec OCT (Sakura) and cryosectioned at 10 μm. Sections were mounted on glass slides, fixed in cold acetone for 20 min, followed by air drying to evaporate acetone completely. Sections were washed in PBS three times and stained with αIgD-PB (Thermo Fisher Scientific 48-5993-82, 1:50), αCD4-PE (Thermo Fisher Scientific 12-0041-82, 1:100), and PNA-FITC (Vector Laboratories FL-1071, 1:400) for 30 min at room temperature. Images were acquired on a LSM 700 system (Carl Zeiss, Inc.).

**Statistical methods**. Statistical tests were performed using Prism 6.0c (GraphPad). Significance was determined by unpaired Student's t-test with a 95% confidence interval. $*p < 0.05$, $**p < 0.01$, $***p < 0.001$, ns not significant.

**Data availability**. The authors declare that the data supporting the findings of this study are available within the article and its Supplementary Information files, or are available upon reasonable requests to the authors. RNA-seq data for naive mature B cells or GC B cells from B-KO mice or WT littermates are available from the NCBI GEO database (GEO GSE113975).

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

## Acknowledgements

This work was supported by NIH grants AI089935, AI103646, AI108651, and AI123782 (L.-F.L.). H.-M.L. is an Irvington Fellow of the Cancer Research Institute. A.Y.R. is an investigator with the Howard Hughes Medical Institute. We thank the technical services provided by the Transgenic Mouse Model Core Facility of the National Core Facility Program for Biotechnology, Ministry of Science and Technology, Taiwan and the Gene Knockout Mouse Core Laboratory of National Taiwan University Center of Genomic Medicine. We thank all members of our laboratory for discussions.

## Author contributions

Conceived and designed the experiments: S.Ch., H.-M.L., and L.-F.L. Performed the experiments: S.Ch., H.-M.L., M.-C.C., L.-L.L., and Y.S.C. Analyzed the data: S.Ch., H.-M.L., Y.S.C., H.-Y.H., S.S.H., A.A.K., S.Cr., and L.-F.L. Contributed reagents/materials/analysis tools: I.-S.Y., H.-Y.H., S.S.H., A.A.K., S.-W.L., and A.Y.R. Wrote the paper: S.Ch., H.-M.L., N.D.A., and L.-F.L.

## Additional information

**Competing interests:** The authors declare no competing interests.

