## [Peer Review File · Nature Communications]

Reviewers' comments:

Reviewer #1 (T/B interaction, germinal center)(Remarks to the Author):

The GC response underlies long-lived humoral immunity and is under elaborate regulation and control, including those exerted by micro-RNA species. Mir-146 is well-established for its immunoregulatory role, and increasing evidence reveals its direct involvement in modulating various aspects of the GC biology. In particular, previous studies have demonstrated that by targeting ICOS, mir-146a helps to limit the magnitude of the GC response. In this paper, Cho and colleagues further explore cell type-specific functions of mir-146a and mir-146b using conditional KO mice. Their results reveal a B cell-intrinsic, non-redundant function of mir-146a in controlling CD40-NF- κ B pathway and regulating the magnitude of GC responses. The authors also provide evidence that mir-146a and mir-146b are redundantly required in the T-cell compartment for GC responses of a normal magnitude to proceed. This latter subtlety is in contrast to previous studies that conclude mir-146a as required in the T-cell compartment for normal GC responses. These results are potentially interesting, although additional analyses are required to firm up some of the crucial conclusions and to enrich the study.

Major concerns:

1. Given that CD21-cre delete mir-146a from all B cells, including marginal zone B cells and those with regulatory functions. The authors should provide evidence to rule in or out potential abnormalities of those non-GC B cells as a contributing factor of the KO phenotype. Potentially, mir-146a deletion by AID-cre may help pinpoint the effect to GC B cells.
2. The functional evidence in vitro that CD40 and associated NF- κ B signaling pathways underlie the B cell-intrinsic effects of mir-146a is strong. However, the in vivo rescue experiment (Fig. 4e) is problematic. The authors only examined a single time point, day 12 post SRBC immunization. This is a rather late time point for SRBC-induced GC responses, leaving it possible that CD40 heterozygosity may not fully correct the exaggerated GC response and additional pathways are involved. Therefore, a time course analysis including earlier and response-peak time points is mandatory, and Tfh cells should also be analyzed and quantified.
3. A related issue in Fig. 4e is that because the authors constructed 1:1 mixed chimera, it is important to normal Ly5.1- GC% against Ly5.1- naïve B cells% for individual chimeric animals in order to control for variation in the reconstitution efficiency. Statistical comparison between the B-KO/CD40 \pm and WT groups should be provided as well. BTW, why is 1.98% chosen to 'represent' the B-KO/CD40 \pm group in the FACS plot in Fig 4e, when the group mean appears ~1.1%?
4. Activation of NF- κ B pathway in GC B cells induces IRF4 and tends to terminate the GC program in favor of plasma cell differentiation. Because CD40-NF κ B pathway is hyperactive in the absence of mir-146a, an increased tendency of plasma cell formation on a per cell basis could also contribute to the KO phenotype (e.g. increased Ab titer). The authors are urged to examine plasma cell output kinetics after immunization.
5. The observation that the mir-146a deficiency in TB-KO mice did not give rise to a more severe phenotype than in the B-KO situation is quite interesting. It seems plausible that mir-146a controls collaborative and mutually dependent aspects in B and T cells. Do the authors think ICOS in T cells and CD40 in B cells are regulating some common pathways? Given that mir-146a controls CD40 dependent pathway in B cells, could it actually control CD40L production in T cells, and redundantly by mir-146b as well? The authors are advised to explore this further to enrich the study.

Minor concerns

1. Do mice fair worse if B cells lack both mir-146a and -146b?

Reviewer #2 (T activation, miRNA)(Remarks to the Author):

The function of miR-146a has been extensively studied. In this manuscript, Lu and colleagues revisited the function of this miRNA in GC formation with more sophisticated genetic systems. Employing B cell-specific, T cell-specific, and T+B cell-specific mir146a genetic deletion, as well as T cell-specific miR-146a plus miR-146b ablation, the authors provided strong evidences and mechanisms that miR-146a expression in B cells is sufficient to restricted excessive GC formation. Mechanistically, this is through a multiple targeting mechanism enriched in CD40 signaling. In addition, against the previous published data, mir146a deletion alone in T cells failed to generate GC phenotype, instead, the amplified GC responses can be observed if both miR-146a and miR-146b are ablated. Although the phenotype is relatively moderate and the knowledge is not completely novel, this manuscript indeed provided solid evidence to conclude the role and mechanism for miR-146 in GC responses, which will greatly benefit the field. It is clearly written and well presented. Only a few minor concerns need to be addressed before the acceptance.

1) The role of miR-146b in T cells. Although in comparison to miR-146a, the expression level of miR-146b is lower in B and T cells, it does not exclude the possibility that miR-146b may play a more important role than miR-146a. The double knockout data cannot address this issue. It is necessary to present T cell-specific mir146b deletion data.

2) Recent data suggested that gene deletion by CD4-Cre is not limited to T cells. Some macrophage and DC populations are also CD4+ and miR-146a could be depleted in these population and the hypoactivation of these innate cells impacts the immunization result. It will strengthen the conclusion if there is data to indicate that miR-146a deletion does not change the antigen presentation, co-stimulation, or cytokine profiles of these innate populations.

3) It is necessary to provide data indicating that mir146a deletion driven by hCD2-Cre does not impact T and B cell development.

We would like to thank reviewers for their insightful and constructive comments, addressing which, we hope, resulted in a significantly improved manuscript. Several new figures, which in our opinion do not substantially help strengthen the central messages of our manuscript, are presented for reviewers only in this letter. Nonetheless, we would be happy to reconstruct our manuscript if the reviewers and the editor deem necessary. (Significant changes within the body of the manuscript are underlined)

Reviewer #1

The GC response underlies long-lived humoral immunity and is under elaborate regulation and control, including those exerted by micro-RNA species. Mir-146 is well-established for its immunoregulatory role, and increasing evidence reveals its direct involvement in modulating various aspects of the GC biology. In particular, previous studies have demonstrated that by targeting ICOS, mir-146a helps to limit the magnitude of the GC response. In this paper, Cho and colleagues further explore cell type-specific functions of mir-146a and mir-146b using conditional KO mice. Their results reveal a B cell-intrinsic, non-redundant function of mir-146a in controlling CD40-NF- κ B pathway and regulating the magnitude of GC responses. The authors also provide evidence that mir-146a and mir-146b are redundantly required in the T-cell compartment for GC responses of a normal magnitude to proceed. This latter subtlety is in contrast to previous studies that conclude mir-146a as required in the T-cell compartment for normal GC responses. These results are potentially interesting, although additional analyses are required to firm up some of the crucial conclusions and to enrich the study.

Major concerns:

1. Given that CD21-cre delete mir-146a from all B cells, including marginal zone B cells and those with regulatory functions. The authors should provide evidence to rule in or out potential abnormalities of those non-GC B cells as a contributing factor of the KO phenotype. Potentially, mir-146a deletion by AID-cre may help pinpoint the effect to GC B cells.

We agree with the reviewer that our CD21-cre mouse model would result in the deletion of miR-146a in all B cell subsets and that using another cre line that renders GC B cell-specific gene targeting could further reveal the role of miR-146a in GC B cells. To address this point, we induced deletion of miR-146a in GC B cells by breeding our *miR-146a^{fl}* mice to C γ 1-cre mice, a mouse model in which 85~95% of GC B cells were shown to undergo cre-mediated recombination upon SRBC immunization ¹. As shown in **Fig.S3**, unlike what has been shown in our B-KO mice, only mild increases in both GC B cell and Tfh cell numbers could be detected in mice with GC B cell-specific miR-146a ablation (GCB-KO) at day 28 post SRBC immunization. These results are interesting, but not entirely surprising. Considering that CD40 signaling was previously shown to act predominately in the initial phase of the GC reaction and is largely dispensable for GC B cell expansion ², our data further support the notion that miR-146a controls GC B cell responses mainly through regulating the CD40 signaling pathway.

2. The functional evidence in vitro that CD40 and associated NF- κ B signaling pathways underlie the B cell-intrinsic effects of mir-146a is strong. However, the in vivo rescue experiment (Fig. 4e) is problematic. The authors only examined a single time point, day 12 post SRBC immunization. This is a rather late time

point for SRBC-induced GC responses, leaving it possible that CD40 heterozygosity may not fully correct the exaggerated GC response and additional pathways are involved. Therefore, a time course analysis including earlier and response-peak time points is mandatory, and Tfh cells should also be analyzed and quantified.

We agree with the reviewer that the time point (day 14) we used in the original submission is later than the peak time for the SRBC-induced GC responses which normally occur at day 10 post immunization. To this end, we have re-conducted our mixed BM chimera studies to include two additional time points (day 5 and day 10) in our revised manuscript. As shown in new **Fig.4f** and **Fig.S5**, in both early and response-peak time points, exaggerated GC responses could also be similarly rescued by reducing CD40 gene dosage. On the other hand, unlike GC B cell responses, we did not detect any significant alteration in Tfh cell numbers even though increased Tfh cell frequencies could be observed in mice with higher ratios of reconstituted KO versus WT compartments (**Fig.R1**). These results suggested that the lack of clear Tfh cell phenotypes in our mixed BM chimeras study is likely due to the dilution effect resulting from the presence of various sizes of Ly5.1+ WT GC B cell populations in the chimeric mice (**Fig.R1**). Moreover, as CD40 heterozygosity is not specifically restricted to B cells, partial deletion of CD40 in other non-B cells (e.g. DCs) in the B-KO/CD40^{+/-} compartment would further confound the interpretation of the observed Tfh cell phenotypes. Taken together, we hope the reviewer can appreciate the limitation of the mixed BM chimera approach in analyzing Tfh cells and would still agree that our newly provided data have sufficiently supported our proposed model in which the miR-146a/CD40 axis plays the major role in regulating GC B cell responses.

3. A related issue in Fig. 4e is that because the authors constructed 1:1 mixed chimera, it is important to normal Ly5.1- GC% against Ly5.1- naïve B cells% for individual chimeric animals in order to control for variation in the reconstitution efficiency. Statistical comparison between the B-KO/CD40+/- and WT groups should be provided as well. BTW, why is 1.98% chosen to 'represent' the B-KO/CD40+/- group in the FACS plot in Fig 4e, when the group mean appears ~1.1%?

To account for the variation in the reconstitution and immunization efficiency in the mixed BM chimera study, we thought it is best to normalize the Ly5.1- GC B cell frequencies against the WT Ly5.1+ GC B cell counterparts in the same chimeric mice in order to obtain a fair comparison between different groups. Therefore, all the data shown in the prism plots in the original Fig.4e were the ratios of Ly5.1- GC B cell frequency versus Ly5.1+ GC B cell frequency rather than the actual frequencies of Ly5.1- GC B cells in each chimeric mice. To this end, in the B-KO/CD40^{+/-} group, the chosen representative FACS plot of Ly5.1- GC B cell frequency is 1.98% and the FACS plot of Ly5.1+ GC B cell frequency in the same chimeric mouse is 1.49%. As such, the ratio of the GC B cell response in this particular mouse is: $1.98/1.49 = 1.33$ which is closer to the group mean (the actual value is 1.22) shown in the prism plot. We apologize for the confusion and not having sufficient information for this figure. Additional description about the mixed BM chimeras results is now included in the revised manuscript.

4. Activation of NF-Kb pathway in GC B cells induces IRF4 and tends to terminate the GC program in favor of plasma cell differentiation. Because CD40-NFkB pathway is hyperactive in the absence of mir-146a, an increased tendency of plasma cell formation on a per cell basis could also contribute to the KO phenotype

(e.g. increased Ab titer). The authors are urged to examine plasma cell output kinetics after immunization.

We thank the reviewer for this helpful suggestion. To address this comment, we examined plasma cell responses upon SRBC immunization. As shown in new **Fig.2d** and **Fig.S2**, we could detect elevated plasma cell frequencies in B-KO mice at day 14 post SRBC immunization consistent with increased SRBC-specific IgG responses observed in those mice (**Fig.2e**).

5. The observation that the mir-146a deficiency in TB-KO mice did not give rise to a more severe phenotype than in the B-KO situation is quite interesting. It seems plausible that mir-146a controls collaborative and mutually dependent aspects in B and T cells. Do the authors think ICOS in T cells and CD40 in B cells are regulating some common pathways? Given that mir-146a controls CD40 dependent pathway in B cells, could it actually control CD40L production in T cells, and redundantly by mir-146b as well? The authors are advised to explore this further to enrich the study.

We thank the reviewer for bringing up this interesting point. As shown in new **Fig.3m**, we can detect miR-146a up-regulation in B cells in response to CD40 stimulation, consistent with the established role of miR-146a functioning as a negative feedback regulator in controlling other signaling pathways^{3, 4}. Interestingly, however, miR-146a expression was not changed in T cells upon ICOS engagement (**Fig.R2**), suggesting that Tfh cells acquire increased expression of miR-146a through activation of ICOS-independent pathways. Moreover, while many genes associated with the CD40 signaling pathway are targeted by miR-146a in B cells, it does not seem to impact the expression of CD40L in T cells. Deletion of miR-146a alone or both family members in T cells did not result in any alteration in CD40L expression (**Fig.R3**). Therefore, while miR-146a serves as a common negative regulator in limiting GC B and Tfh cells during humoral immune responses, it could also target or be regulated by distinct biological pathways in these two immune cell subsets.

Minor concerns

1. Do mice fair worse if B cells lack both mir-146a and -146b?

As shown in the original Fig.S5a (new **Fig.S8a**), B cells, when compared to T cells, express even lower amounts of miR-146b. Therefore, we did not expect to see a similar impact on B cells upon deletion of both miR-146a and b compared to what was shown in mice with T cell-specific deletion. Nevertheless, to formally address this comment, we generated mice with B cell-specific ablation of both miR-146 family members (B-DKO). As shown in **Fig.R4**, upon SRBC immunization, B-DKO mice only harbored minimal increases in GC B cell frequencies compared to what was detected in B-KO mice, confirming the proposed major role of miR-146a in regulating GC B cell responses..

Reviewer #2

The function of miR-146a has been extensively studied. In this manuscript, Lu and colleagues revisited the function of this miRNA in GC formation with more sophisticated genetic systems. Employing B cell- specific,

T cell-specific, and T+B cell-specific mir146a genetic deletion, as well as T cell-specific miR-146a plus miR-146b ablation, the authors provided strong evidences and mechanisms that miR-146a expression in B cells is sufficient to restricted excessive GC formation. Mechanistically, this is through a multiple targeting mechanism enriched in CD40 signaling. In addition, against the previous published data, mir146a deletion alone in T cells failed to generate GC phenotype, instead, the amplified GC responses can be observed if both miR-146a and miR-146b are ablated. Although the phenotype is relatively moderate and the knowledge is not completely novel, this manuscript indeed provided solid evidence to conclude the role and mechanism for miR-146 in GC responses, which will greatly benefit the field. It is clearly written and well presented. Only a few minor concerns need to be addressed before the acceptance.

We thank the reviewer for the overall positive comments on our manuscript.

1. The role of miR-146b in T cells. Although in comparison to miR-146a, the expression level of miR-146b is lower in B and T cells, it does not exclude the possibility that miR-146b may play a more important role than miR-146a. The double knockout data cannot address this issue. It is necessary to present T cell-specific mir146b deletion data.

To date, there has been no report clearly demonstrating that miR-146a and miR-146b could exhibit different regulatory function. Nevertheless, we could not completely exclude the possibility that miR-146b albeit expressed at a lower level in T cells could potentially be the major player in regulating the Tfh cell responses. As the reviewer alluded to, in this given scenario, deletion of miR-146b in T cells alone might be sufficient to cause Tfh cell phenotypes. To address this comment, we have now included the analysis of mice with T cell-specific miR-146b ablation (T-bKO) in the revised manuscript (**Fig.S9**). To this end, similar to what was reported in mice with T cell-specific miR-146a ablation, T-bKO mice exhibited comparable Tfh and GC B cell responses and did not show any sign of abnormality. Together with our newly provided data obtained from B-DKO mice (which demonstrated a rather minimal impact when miR-146b together with miR-146a were deleted in B cells; please see **Fig.R4** and our response to Reviewer 1's minor critique), we hope the reviewer will agree that it is unlikely that miR-146b plays a more important role than miR-146a in our studies.

2. Recent data suggested that gene deletion by CD4-Cre is not limited to T cells. Some macrophage and DC populations are also CD4+ and miR-146a could be depleted in these population and the hypoactivation of these innate cells impacts the immunization result. It will strengthen the conclusion if there is data to indicate that miR-146a deletion does not change the antigen presentation, co-stimulation, or cytokine profiles of these innate populations.

Since the role of miR-146a as a negative regulator in macrophages and DCs has been extensively studied in the past ^{5, 6, 7, 8}, one would expect to see enhanced rather than attenuated activation in those CD4⁺ innate cell populations if miR-146a was deleted when bred to CD4-cre mice. Nevertheless, to address this comment, we examined the antigen presentation, co-stimulation and cytokine production capacity of these innate populations as suggested by the reviewer. As shown in **Fig.R5**, we have not found any difference in any of the aforementioned biological functions tested. These results strongly suggested that hypoactivation of these innate immune populations is unlikely responsible for the lack of enhanced Tfh cell phenotype in

CD4-cre miR-146a^{fl/fl} mice.

3. It is necessary to provide data indicating that mir146a deletion driven by hCD2-Cre does not impact T and B cell development.

To ensure hCD2-cre driven miR-146a deletion did not impact T and B cell development, we examined different T and B cell subsets in thymus and BM, respectively. As shown in **Fig.R6**, we did not detect any significant difference in any of the developing T and B lymphocyte subsets. These results suggested that the enhanced GC B and Tfh cell phenotypes observed in those mice were not likely resulted from disturbed T and B cell development.

References

1. Casola S, Cattoretti G, Uyttersprot N, Koralov SB, Seagal J, Hao Z, et al. Tracking germinal center B cells expressing germ-line immunoglobulin gamma1 transcripts by conditional gene targeting. *Proceedings of the National Academy of Sciences of the United States of America* 2006, **103**(19): 7396-7401.
2. Basso K, Klein U, Niu H, Stolovitzky GA, Tu Y, Califano A, et al. Tracking CD40 signaling during germinal center development. *Blood* 2004, **104**(13): 4088-4096.
3. Taganov KD, Boldin MP, Chang KJ, Baltimore D. NF-kappaB-dependent induction of microRNA miR-146, an inhibitor targeted to signaling proteins of innate immune responses. *Proceedings of the National Academy of Sciences of the United States of America* 2006, **103**(33): 12481-12486.
4. Lu LF, Boldin MP, Chaudhry A, Lin LL, Taganov KD, Hanada T, et al. Function of miR-146a in controlling Treg cell-mediated regulation of Th1 responses. *Cell* 2010, **142**(6): 914-929.
5. Boldin MP, Taganov KD, Rao DS, Yang L, Zhao JL, Kalwani M, et al. miR-146a is a significant brake on autoimmunity, myeloproliferation, and cancer in mice. *The Journal of experimental medicine* 2011, **208**(6): 1189-1201.
6. Mann M, Mehta A, Zhao JL, Lee K, Marinov GK, Garcia-Flores Y, et al. An NF-kappaB-microRNA regulatory network tunes macrophage inflammatory responses. *Nature communications* 2017, **8**(1): 851.
7. Park H, Huang X, Lu C, Cairo MS, Zhou X. MicroRNA-146a and microRNA-146b regulate human dendritic cell apoptosis and cytokine production by targeting TRAF6 and IRAK1 proteins. *The Journal of biological chemistry* 2015, **290**(5): 2831-2841.
8. Stickel N, Hanke K, Marschner D, Prinz G, Kohler M, Melchinger W, et al. MicroRNA-146a reduces MHC-II expression via targeting JAK/STAT signaling in dendritic cells after stem cell transplantation. *Leukemia* 2017, **31**(12): 2732-2741.

Figure R1. Variation in the reconstitution efficiency in mixed BM chimeras influenced the magnitude of Tfh cell responses. (a) Frequencies of total Tfh cells in mixed BM chimeras at day 14 after SRBC immunization. FACS analysis of (b) total CXCR5⁺PD-1⁺ Tfh cells as well as (c) Ly5.1⁺ and Ly5.1⁻ B220⁺GL7⁺PNA⁺ GC B cell populations in indicated chimeric mice. The data are representative of three independent experiments. Each symbol represents an individual mouse, and the bar represents the mean.

Figure R2. ICOS engagement did not lead miR-146a up-regulation. Quantitative PCR of miR-146a levels in naïve T cells (Tn) upon ICOS stimulation at indicated time points. The data are shown as mean \pm SD and are representative of two independent experiments (n=2-4).

Figure R3. Deletion of miR-146 family members in T cells did not impact CD40L expression. (a) FACS analyses of the expression of CD40L in CD44^{hi}CD4⁺ T cells in spleen from T-KO and T-DKO mice compared to their corresponding WT littermates. **(b)** Fold changes (on the basis of corresponding WT controls) of CD40L mean fluorescence intensity (MFI) were shown. Data represent the mean \pm SD and are representative of 2 independent experiments (n=3-4).

Figure R4. Simultaneously deletion of both miR-146 members in B cells did not lead to further enhanced GC B cell responses. (a) FACS analysis of PNA⁺GL7⁺ GC B cells from spleen of ~8 wks old mice with B cell-specific ablation of both miR-146a and b (B-DKO) or their WT littermates 14 days after SRBC immunization. (b) Fold change (on the basis of corresponding WT controls) of GC B cell frequencies in B-KO and B-DKO mice were shown. Data represent the mean ± SD and are representative of 2 independent experiments (n=3-6). *p<0.05, **p<0.01

Figure R5. CD4-cre-mediated miR-146a deletion in the innate populations did not result in any detectable functional alterations. (a) FACS analysis of MHC II and co-stimulatory molecules (i.e. CD80 and CD86) in DCs or macrophages after LPS(100ng/ml) stimulation for 6hr. (b) OTII T cells were co-cultured with DC isolated from spleens of T-KO mice or their littermates in the presence of OVA for 3 days. Proliferation of OTII T cells was assessed by CFSE dilution. FACS analysis and frequencies of (c) TNF- α and IL-12 in DC, (d) TNF- α , IL-12 and (e) IL-6 in macrophages. Data represent the mean \pm SD and are representative of 2 independent experiments. Each symbol represents an individual mouse.

Figure R6. Mice with hCD2-cre driven miR-146a ablation exhibited normal B and T cell development. FACS analysis and frequencies of (a) B220⁺IgD⁻IgM⁻CD43⁺ Pro- or B220⁺IgD⁻IgM⁻CD43⁻ Pre-B cells, (b) B220⁺IgD⁻IgM⁺CD43⁻ immature or B220⁺IgD⁺IgM⁺CD43⁻ mature B cells in BM and (c) different thymocyte subsets (DN, DP, CD4SP and CD8SP) in thymus from TB-KO mice or their littermate. Data represent the mean ± SD and are representative of 3 independent experiments. Each symbol represents an individual mouse.

Reviewers' comments:

Reviewer #1 (Remarks to the Author):

The authors have largely addressed my previous questions. My remaining concerns are as follow:

1. Given the central importance of CD40 pathway in their model, Bulk analyses by Western blotting in Figures 3d and 4a would not be sufficient. It is important to at least document surface CD40 expression levels in WT and mir-146 KO B cells at the naïve, activated and GC stage. It is also appropriate to document CD40 surface levels in B-KO/CD40+/- cells.
2. In terms of the mixed BM chimera experiment (Figures 4d-f), it is important to show the ratio of Ly5.1-/Ly5.2+ in the naïve or non-GC or total B-cell compartment is actually ~1 across all the groups. The authors should already have the data. This is to ensure the increased ratio in the B-KO group is not due to reconstitution variation in that particular group and/or any potential effects of mir-146 on B-cell development/survival/death at the steady state. This is what was asked in the previous review.
3. More detailed technical information should be provided as to exactly what cells are used for RNA-seq and Western blotting in Figure 3. Which data are from naïve B cells and which from GC B cells? If GC B cells, what is the immunization scheme and time point or they are of the spontaneous type?

Reviewer #2 (Remarks to the Author):

All my previous concerns have been adequately addressed. I would like to recommend it for publication.

We would like to thank reviewers again for their helpful comments. We hope our responses would be able to address most if not all the remaining concerns. (New changes within the body of the manuscript are underlined)

Reviewer #1

The authors have largely addressed my previous questions. My remaining concerns are as follow:

1. Given the central importance of CD40 pathway in their model, Bulk analyses by Western blotting in Figures 3d and 4a would not be sufficient. It is important to at least document surface CD40 expression levels in WT and mir-146 KO B cells at the naïve, activated and GC stage. It is also appropriate to document CD40 surface levels in B-KO/CD40+/- cells.

As shown in our manuscript, rather than directly targeting CD40 itself, miR-146a limits GC B cell responses through controlling many CD40 downstream targets. Upregulation of CD40 is only secondary to the enhanced CD40 signaling. Nevertheless, we agreed with the reviewer that it would be nice to see the difference in surface CD40 expression. With that being said, it has been well established that CD40 signaling induces rapid endocytosis of CD40 and internalization of CD40 is a way to negatively regulate its signaling (Chen et al., 2006, Menard et al., 2016). Since miR-146a functions as a negative regulator in controlling CD40 signaling, the effect of miR-146a deficiency on surface CD40 expression could not be clearly detected as enhanced CD40 signaling in KO B cells would likely lead to increased CD40 internalization (**Fig. R1**). Therefore, we believe that demonstration of increased amounts of total CD40 by our Western blotting studies would more truthfully reflect the consequence of enhanced CD40 signaling resulted from miR-146a deficiency. We understand that despite our efforts we could not answer exactly what the reviewer was asking regarding to showing the difference in surface CD40 expression due to the aforementioned limitation. With that being said, together with our results obtained from CD40 heterozygosity rescuing studies both *in vitro* and *in vivo*, we sincerely hope that the reviewer would agree that we have provided sufficient support to our conclusion that miR-146a controls GC B cell responses mainly through regulating the CD40 signaling pathway.

Fig R1. miR-146a deficiency only led to minimal differences in surface CD40 expression. FACS analysis of the surface expression of CD40 on B220+ B cells in spleen from B-KO and WT control mice.

2. In terms of the mixed BM chimera experiment (Figures 4d-f), it is important to show the ratio of Ly5.1-/Ly5.2+ in the naïve or non-GC or total B-cell compartment is actually ~1 across all the groups. The authors should already have the data. This is to ensure the increased ratio in the B-KO group is not due to reconstitution variation in that particular group and/or any potential effects of mir-146 on B-cell development/survival/death at the steady state. This is what was asked in the previous review.

We examined the ratios of Ly5.1-/Ly5.2+ in total B cell compartment as the reviewer requested. As shown in new **Fig.S6**, we did not detect any difference in total B cell compartment between three chimeric groups during the course of SRBC immunization as the Ly5.1-/Ly5.2+ ratios were ~1 across all three groups as the reviewer suggested. These results further demonstrated that miR-146a deficiency in B cells led to stronger GC B cell responses without impacting the size of total B cell population.

3. More detailed technical information should be provided as to exactly what cells are used for RNA-seq and Western blotting in Figure 3. Which data are from naïve B cells and which from GC B cells? If GC B cells, what is the immunization scheme and time point or they are of the spontaneous type?

We apologize for not listing the technical information on experiment designs clearly. Additional descriptions regarding the cell types as well as the experimental procedure are now included in the main text as well as the material and method section of the final revised manuscript.

REVIEWERS' COMMENTS:

Reviewer #1 (Remarks to the Author):

This reviewer does not believe the authors must show an increase or decrease in CD40 surface expression in BKO cells in order for their model to be correct. However, it is also an obviously simple and rational experiment, particularly given the Western blotting data, and the authors should certainly have had the results. Given the circumstance, the authors should at least mention their surface CD40 staining data, possibly as unpublished observations in the Discussion section, and explain their reasoning as argued in the rebuttal. This will provide expert readers of the paper a chance to better understand the nuance and subtleties.

We would like to thank the reviewer once again for his/her helpful comment. We hope our response has now satisfactorily addressed the remaining concern.

Reviewer #1

This reviewer does not believe the authors must show an increase or decrease in CD40 surface expression in BKO cells in order for their model to be correct. However, it is also an obviously simple and rational experiment, particularly given the Western blotting data, and the authors should certainly have had the results. Given the circumstance, the authors should at least mention their surface CD40 staining data, possibly as unpublished observations in the Discussion section, and explain their reasoning as argued in the rebuttal. This will provide expert readers of the paper a chance to better understand the nuance and subtleties.

We would like to thank the reviewer for agreeing that whether or not a clear increase in surface CD40 expression in B-KO B cells could be detected is not necessary to support our conclusion that miR-146a controls GC B cell responses mainly through regulating the CD40 signaling pathway. With that being said, a paragraph of the description and the explanation of the minimal difference in surface CD40 expression between WT and B-KO B cells is now included in the Discussion section as the reviewer suggested.